# Upstream GPS Vertical Displacement and its Standardization for Mekong River Basin Surface Runoff Reconstruction and Estimation

**Hok Sum Fok [1,2,3,]\*, Linghao Zhou [1,2], Yongxin Liu [4,5], Zhongtian Ma [1,2] and Yutong Chen [1,2]**

1   School of Geodesy and Geomatics, Wuhan University, Wuhan 430079, China;
    lhzhou2016@whu.edu.cn (L.Z.); zt__ma@whu.edu.cn (Z.M.); ytchen2016@whu.edu.cn (Y.C.)
2   Key Laboratory of Geospace Environment and Geodesy, Ministry of Education, Wuhan University,
    Wuhan 430079, China
3   Institute of Marine Science and Technology, Wuhan University, Wuhan 430079, China
4   School of Earth and Space Sciences, Peking University, Beijing 100871, China; yxliugeo@pku.edu.cn
5   Engineering Research Center of Earth Observation and Navigation (CEON), Ministry of Education of the
    PRC, No. 5 Yiheyuan Road, Haidian District, Beijing 100871, China
\*   Correspondence: xshhuo@sgg.whu.edu.cn; Tel.: +86-027-6877-8649

**Abstract:** Surface runoff ($R$), which is another expression for river water discharge of a river basin, is a critical measurement for regional water cycles. Over the past two decades, river water discharge has been widely investigated, which is based on remotely sensed hydraulic and hydrological variables as well as indices. This study aims to demonstrate the potential of upstream global positioning system (GPS) vertical displacement (VD) and its standardization to statistically derive $R$ time series, which has not been reported in recent literature. The correlation between the in situ $R$ at estuaries and averaged GPS-VD and its standardization in the river basin upstream on a monthly temporal scale of the Mekong River Basin (MRB) is examined. It was found that the reconstructed $R$ time series from the latter agrees with and yields a similar performance to that from the terrestrial water storage based on gravimetric satellite (i.e., Gravity Recovery and Climate Experiment (GRACE)) and traditional remote sensing data. The reconstructed $R$ time series from the standardized GPS-VD was found to have a 2–7% accuracy increase against those without standardization. On the other hand, it is comparable to data that are obtained by the Palmer drought severity index (PDSI). Similar accuracies are exhibited by the estimated $R$ when externally validated through another station location with in situ time series. The comparison of the estimated $R$ at the entrance of river delta against that at the estuaries indicates a 1–3% relative error induced by the residual ocean tidal effect at the estuary. The reconstructed $R$ from the standardized GPS-VD yields the lowest total relative error of less than 9% when accounting for the main upstream area of the MRB. The remaining errors may be the result of the combined effect of the proposed methodology, remaining environmental signals in the data time series, and potential time lag (less than a month) between the upstream MRB and estuary.

**Keywords:** Runoff; GPS; GRACE satellite gravimetry; Mekong River Basin; PDSI

## 1. Introduction

River water discharge (RWD) is among the critical hydrological components of river basins measured near the mouth of estuaries [1,2]. Another representation of the RWD at the estuaries is surface runoff ($R$), in which it is the RWD divided by the total basin area. In order to prepare for unpredictable losses in agricultural products and economy (e.g., [3–6]), the continuous monitoring of RWDs is necessary for tracking abrupt hydrological changes (i.e., droughts and floods) in deltaic

environments. There is no global coverage of gauging network, however, that monitors the RWDs [7]. Apart from this, the frequency of discharge data acquisition has continuously declined since the late 1970s [8] because of insufficient funds for facility maintenance and upgrade [9]. As a result, indirect methods for the RWD monitoring, such as remote sensing (RS), have recently gained increasing interest.

Traditional RS, such as Landsat Thematic Mapper (TM) and its Enhanced TM Plus (ETM+) images, as well as moderate resolution imaging spectrometer (MODIS), have passively recorded instantaneous surface parameters since the 1990s (e.g., [7]). The surface parameters obtained from RS [10–13], such as flood area inundation, land surface temperature (LST), normalized difference vegetation index (NDVI), and RS-derived geometric variables (e.g., river width), allow the direct correlation with water level or RWD. Except for the RS-derived geometric variables, the foregoing localized RS data yield indirect relationships to the RWD. Although RS-derived geometric variables can also be employed as inputs to infer the RWD through Manning's equation and its modified form (e.g., [14–21]), the accuracy of the estimated RWD is region-dependent because of the technique's ability to detect changes in rivers with short widths [22] and the regional availability of roughness coefficients [23,24].

More recently, space geodetic techniques, such as satellite radar altimetry and the Gravity Recovery and Climate Experiment (GRACE), have extensively been utilized to correlate with the RWD (e.g., [25,26]). These observations, hereafter, are referred to as space geodetic-observed variables. Satellite radar altimetry is able to directly monitor water level variations over water bodies, such as river and lakes (e.g., [27,28]). The water level relates to the RWD via a power function (e.g., [29,30]); hence, the time series of RS RWD is derived by directly correlating the measured satellite altimetric water level with in situ RWD measurements (e.g., [31,32]). The use of a basin-wide RWD estimation that employs multisatellite altimetric water level data has also been demonstrated in the Mekong River Basin (MRB) [25]. The reflected signal contains radar altimetry footprints are contaminated by land surfaces, however, when the river width is smaller than 5 km, such as that found in [33], thereby significantly lowering their accuracy near the riverbank.

Although satellite radar altimetry can actively record along-track surface oscillations of inland water bodies (e.g., [34]), GRACE can directly measure time-variable gravity, thereby making it possible to calculate terrestrial water storage fluctuations at a global or regional scale (e.g., [35,36]). The terrestrial water storage, being one of the water balance components, can physically relate to the RWD (e.g., [26,37]). Therefore, the RWD can be estimated from the terrestrial water storage. GRACE-inferred terrestrial water storage (hereinafter called GRACE-S) also allows the calculation of solid Earth vertical surface deformation (e.g., [38–41]) that is consistent with the recorded global positioning system (GPS) vertical displacement (VD) (hereinafter called GPS-VD) (e.g., [42–44]), as demonstrated and validated in different geographic regions [45–47]. Conversely, the GPS-VD can also be utilized to infer the terrestrial water storage (e.g., [48–51]). Given the foregoing GRACE-inferred physical quantity that is comparable to the GPS-VD, it is anticipated that the latter can be a potential alternative for reconstructing the RWD; no paper in this regard has ever been published in recent literature.

Essentially, the GRACE-S and its standardization have been recently demonstrated to have a good correlation with water level [52] and *R* [53]. The reason for the standardization is that it enhances the regional characteristics of the averaged time series [54] when local means and variances are largely different from the regional one [55], thus mitigating systematic influences due to geographic environment [56]. Given the aforementioned similarity between GRACE-inferred quantities and GPS-VD observations, the GPS-VD and its standardization can presumably achieve a similar quality to that of GRACE in capturing the time series of *R* and standardized *R* via correlative analysis, respectively.

The Mekong River Delta (MRD), a geographic region that is vital for food (e.g., [57]) and water security (e.g., [58]) in Southeast Asia, is the MRB downstream region immediately before the freshwater is discharged into the coastal ocean. The *R* of the MRD is affected by dam operation, which increases (decreases) the discharge flow during drought (flooding) seasons under the principle of no significant annual changes [59,60]. Regardless of whether the dam is in the upstream or downstream MRB, the effects of all dam operations accumulate. The accumulated effects that generate biases should be

partly systematic for any specific month, year-on-year [61]. These biases can be partly reduced by the subtraction (or differencing) process in the above standardization procedure. The aforementioned reasons justify the potential use of the GPS-VD standardization that is obtained at the upstream of the MRB to correlate with the *R* in the MRD statistically.

The potential use of the GPS-VD and its standardization is explored in the upstream of the MRB for statistically correlating with the *R* at a hydrological station in the MRD on a monthly scale via linear regression. The fitted parameters are then utilized to estimate the time series of *R* of another location having an in situ time series for external validation. The available RS instantaneous data (NDVI [62] and LST [63]), Palmer drought severity index (PDSI) [64], and GRACE-S and its standardization [65], are employed for the purpose of comparison.

The structure of this paper is arranged as below: Section 2 describes the MRB and MRD geography; Section 3 illustrates the datasets and their processing; Section 4 presents the methodology and evaluation metrics; Section 5 demonstrates the reconstruction and estimation of *R* based on the GPS-VD and its standardized form, while compared to those based on the NDVI, LST, PDSI, and GRACE-S and its standardization; Section 6 summarizes the conclusions.

## 2. Geographic Environment of the Mekong River Basin and Mekong River Delta

The Mekong River, originating from the three-river headwater region in the eastern Qinghai–Tibetan Plateau, is the transboundary river across the Southeast Asian continent [66]. Water first flows through the Lancang River in Yunnan Province within the boundary of China, followed by Laos, Myanmar, Thailand, Cambodia, and Vietnam. Spanning 25° of latitude, the total surface area of the entire MRB is around 795,000 km$^2$ [67] (Figure 1).

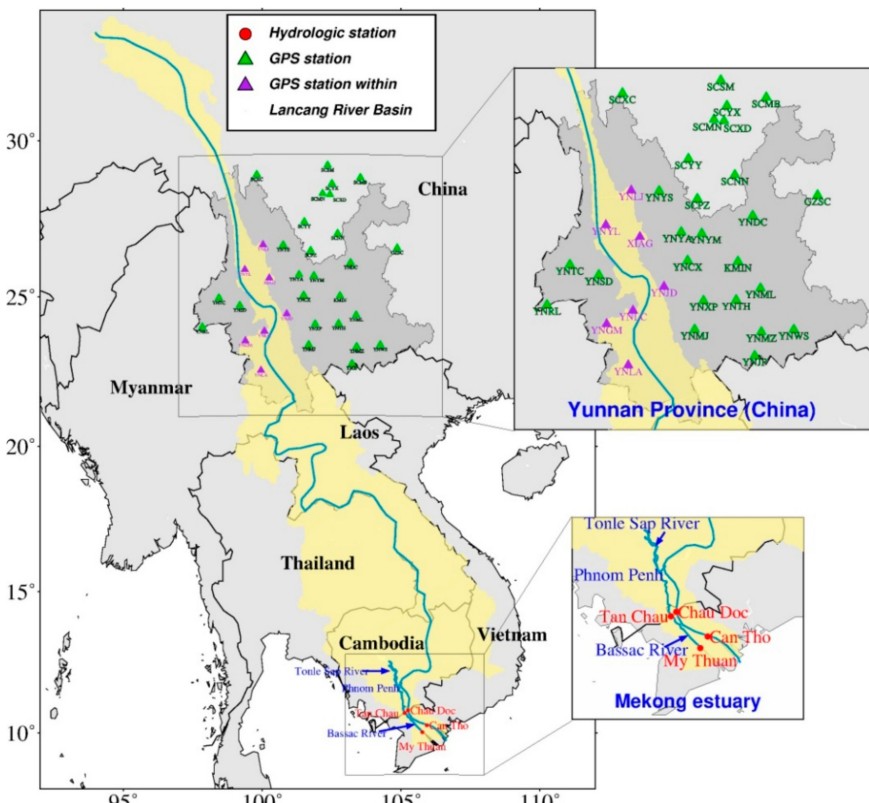

**Figure 1.** Mekong River Basin boundary with hydrological stations (i.e., My Thuan, Can Tho, Tan Chau, and Chau Doc stations) located in the Mekong River Delta and with GPS stations covering the entire Yunnan province and Lancang River Basin within Yunnan province.

Situated at latitude 21°–29°N and longitude 94°–102°E, Lancang River within Yunnan Province, China (hereinafter abbreviated as LRWY), constitutes the main component of the upstream MRB with an altitude ranging from 1500 to 4000 m and descending from north to south [68,69]. It is climate-driven and affected by the Indian monsoon [70]; its distinct rainy and dry seasons are the principal seasonal characteristics. During the wet season (May–October), the SW monsoons from the Bay of Bengal yield the almost entire annual rainfall. During the dry season (November–April), however, severe drought may occur [71]. Bare karst geology, which is characterized by rocks with low permeability, surrounds rocks with high permeability at the eastern part of Yunnan Province. This further causes rapid water infiltration into the ground, aggravating drought conditions [72].

The hydrological extremes in the upstream area significantly affect agriculture, living environments, and economy in the midstream and downstream areas where half a billion people live within China and the country's transboundary [73]. In addition, the constructed massive dams have raised water conflicts among various countries in Southeast Asia [74], despite the aim to regulate flow during extreme hydrological periods [59,60]. The dam operated at different times of the year modifies the upstream *R*, which would adversely impact the water availability downstream (e.g., [75,76]). This indicates that understanding the hydrologically related variables upstream is necessary in order to raise an early alert of extreme events that may occur in the downstream MRB, in particular, the MRD.

The MRD is a transition zone that is seasonally affected by both water discharge of the MRB, and ocean tidal processes propagated landward [57] via the Bassac River and Mekong River branches [77,78]. In addition, the regulation effect of Cambodia's Tonle Sap Lake on the total discharge of the MRB is substantial (e.g., [79–81]) before discharging into the northern part of Sunda Shelf.

## 3. Datasets and Their Processing

### 3.1. In Situ Discharge and Passive Remote Sensing Data

Given the regulated effect of Tonle Sap Lake and the backwater effect of ocean tides that govern the total discharge of the MRB, the selection of in situ stations near the estuary mouth is a critical task. Both aforementioned effects are to be minimized in the selected stations. The station at Phnom Penh, despite far away from the mouth of estuaries, is not selected because it intersects with several tributaries that would significantly modify the overall temporal discharge pattern. In this study, the Tan Chau and Chau Doc stations, located at the entrance of the MRD and interior limit of the transition zone [78], are chosen where both the abovementioned effects are minimized (Figure 1). Despite being closer to the mouth of estuaries, the Can Tho and My Thuan stations are also employed to assess the backwater impact caused by ocean tides on the *R* time series estimation.

The station discharge time series obtained from the Mekong River Commission (MRC) is available at http://www.mrcmekong.org. The selected WD time series data spanning from January 2012 to December 2014 are extracted because of the shorter time span of GPS data. Since half-daily and daily period of ocean tides are the most dominant ocean tidal forcing, the time series of Tan Chau and Chau Doc pair of stations (hereinafter called TC-pair station) and that of My Thuan and Can Tho pair of stations (hereinafter called MC-pair station) are summed up, respectively, serving as a further monthly averaging process to further mitigate the ocean tidal effect.

The observed RWD (in m$^3$/s), accounting for converting second into day, meter into millimeter and dividing by the total basin area (i.e., 795,000 km$^2$), can then be converted into a daily *R* (in mm/day). The *R* at a monthly scale (in mm/month) is then computed by adding daily *R* together. Both the TC-pair and MC-pair station time series exhibit similar temporal patterns (Figure 2).

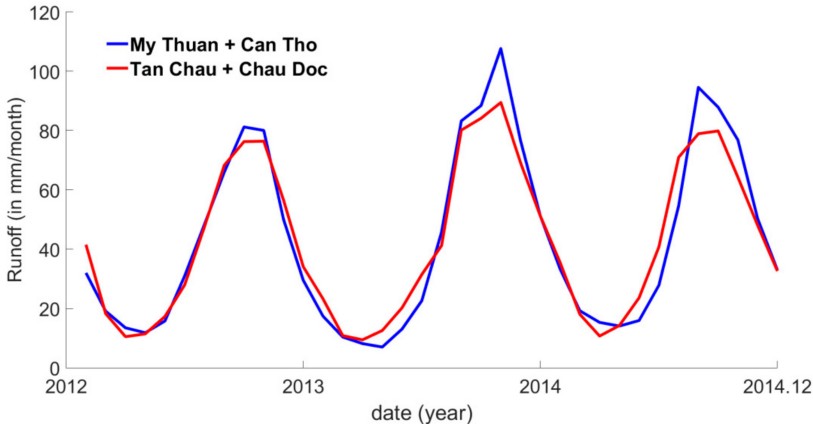

**Figure 2.** Runoff time series of MC-pair station, and TC-pair station.

The LST from MOD11C3 and NDVI from MOD13C2 MODIS products are the traditional RS data that can be downloaded at the Land Processes Distribution Active Archive Center (https://lpdaac.usgs.gov/dataset_discovery/modis/modis_products_table). Both datasets are directly employed to compare against the GPS-reconstructed *R* time series.

### 3.2. Palmer Drought Severity Index

The PDSI [64], downloaded at http://www.cgd.ucar.edu/cas/catalog/climind/pdsi.html [82], is a widely employed index to quantify meteorological drought using worldwide precipitation and temperature data time series to model relative dryness. The index ranges from −10 (dry) to +10 (wet) with a 2.5°×2.5° spatial resolution.

### 3.3. GRACE Terrestrial Water Storage and Its Standardization

Five monthly GRACE solution data products are employed, including the Center for Space Research (CSR) Release (RL) 05 (hereinafter abbreviated as CSR RL05), RL06 (hereinafter abbreviated as CSR RL06), and its RL06 mascon solution (hereinafter abbreviated as CSR-mascon), Jet Propulsion Laboratory (JPL) RL 05 (hereinafter abbreviated as JPL RL05), and GeoforschungsZentrum (GFZ) RL 05 (hereinafter abbreviated as GFZ RL05). Except for the monthly CSR-mascon GRACE-S readily available at a 0.25°×0.25° grid that can be downloaded at http://www2.csr.utexas.edu/grace/RL06_mascons.html, all other monthly GRACE Level-2 data represent mass changes in terms of Stokes coefficients (SC), which can be downloaded at http://icgem.gfz-potsdam.de/series.

While the SC of the JPL RL 05 are expanded up to degree 90, the SC of the CSR RL05, CSR RL06, and GFZ RL05 are expanded up to degree 60. Using equations in [83], the SC of the CSR RL05, CSR RL06, GFZ RL05 and JPL RL05 can be converted into GRACE-S that is interpolated into a 1°×1°grid.

Except for the CSR-mascon, before converting SC into a GRACE-S time series, the degree-one and $C_{20}$ terms in SC are added and replaced, respectively, to correct the geocenter motion and the geoid [84,85]. In addition, a de-striping procedure is applied. As tested in this study, a Gaussian filter with a 350-km radius is the optimal radius chosen to reduce the uncertainties arising from correlated errors of TWS data in space at finer resolutions [86,87].

The processed monthly GRACE-S are then used to reconstruct *R* (i) by directly correlating with the in situ *R* and (ii) by calculating the recently proposed drought index based on GRACE (hereinafter abbreviated as GRACE-SI) [65], in which it is subsequently correlated with the standardized in situ *R*.

To reduce GRACE-S anomalies, the median of GRACE-S for each month is employed to calculate GRACE-SI [52] as follows:

$$SI_{i,j} = \frac{S_{i,j} - median(S_j)}{s_j} \tag{1}$$

where $SI_{i,j}$ and $S_{i,j}$ are the GRACE-SI and GRACE-S for month $j$ of each year $i$, respectively; $S_j$ and $s_j$ are GRACE-S and its corresponding sampled standard deviation, respectively. The same calculation procedure for standardizing in situ $R$ is performed using Equation (1).

### 3.4. GPS Vertical Displacement and Its Standardization

To obtain a daily GPS-VD in the upstream of the MRB, GAMIT FORTRAN software [88] is employed to preprocess the raw observations of 33 GPS stations that monitor Yunnan Province. The observations during 2012–2014 are available from the Crustal Movement Observation Network of China (CMONOC). This is performed by a stochastically constrained network solution (i.e., assigned with a 5-cm uncertainty) [89] aligned to 24 IGS stations in the ITRF2008 coordinate reference frame that surrounds China [90]. The Earth orientation parameters are also constrained to a priori values listed in the International Earth Rotation Service (IERS) Bulletin B.

Standard correction procedures are applied (i) to constrain the orbits to the IGS final ephemeris; (ii) to make corrections for the first three-order terms of the ionospheric delay in GAMIT [91]; (iii) to make corrections for the tropospheric delay using Vienna mapping function 1 [92] and the global modelof pressure and temperature applied in Geodesy [93]; (iv) to apply the antenna offsets given by the IGS filed antenna correction; (v) to remove the non-tidal atmospheric loading using the MIT correction data files; (vi) to remove the ocean tidal loading by choosing the FES2004 model [94] option in GAMIT [88]; (vii) to remove tides due to the solid Earth and pole according to the IERS standard [95]. Moreover, to yield a purely hydrological signal similar to that of GRACE, vertical crustal displacement caused by ocean bottom pressure changes in the GPS-VD should be corrected using half the daily-modeled data downloaded at the Global Geophysical Fluid Center (http://geophy.uni.lu/).

Outliers that are larger than twice the standard deviation are discarded. To reduce the aliasing effect and draconitic errors (e.g., ~351 d [96,97]) in the seasonal signal, spectral filtering is applied after the fast Fourier transformation (FFT) of the GPS-VD time series. After the spectral filtering, the first peak is apparently closer to 1 cpy (Figure 3). The 33 filtered GPS spectra are then transformed back into the time domain via the inverse FFT. In general, the height extremes of the GPS time series are reduced after filtering.

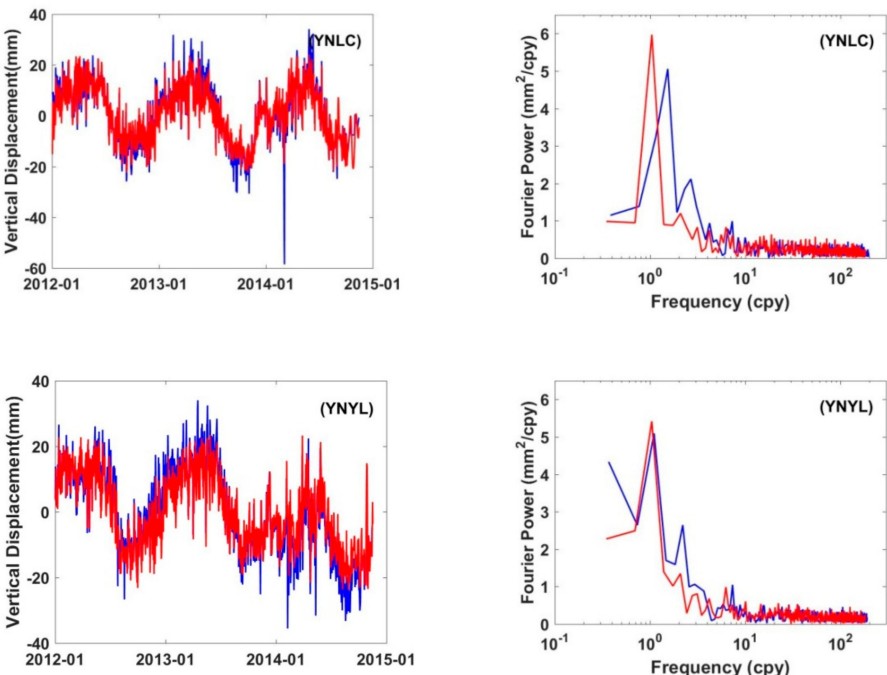

**Figure 3.** Unfiltered (blue) and filtered (red) GPS time series and their respective Fourier power spectra for two selected GPS stations (YNLC and YNYL).

Explored at a monthly scale, the daily GPS-VD are averaged every month to yield the monthly VD. To obtain the monthly standardized GPS-VD data, the same calculation procedure is performed using Equation (1).

## 4. Methodology and Assessment Metrics

### 4.1. Correlative Analysis and Runoff Standardization

In this study, all RS and space geodetic-observed variable data are averaged separately over the bounded square covering Yunnan Province and LRWY (Figure 1) before reconstruction and estimation of *R* in the MRD. Note that smoothing is applied to all the data before the correlative analysis.

The *R* reconstruction is achieved by a direct linear model fitting with a constant offset, *b*, and a slope, *a*, which is given by:

$$y_t = ax_t + b \tag{2}$$

where $y_t$ and $x_t$ are the in situ *R* (or standardized *R*) at the selected stations (MC-pair and TC-pair) and remotely sensed variables (or its standardized form or indices) at monthly epoch *t*, except a one-month time shift was applied to NDVI and LST to improve the reconstruction performance. In addition, the GPS-VD, being negatively correlated with GRACE-S and in situ *R*, was multiplied by a negative one. Note that *a* and *b* are the parameters to be estimated during the model fitting process.

The reconstructed *R* when quantitatively evaluated against the same in situ *R* used for the reconstruction are referred to as internal evaluation, whereas the above estimated parameters determined from the *R* reconstruction that are subsequently used for the *R* estimation and evaluated against at other stations in the MRD are referred to as external evaluation; both evaluations assess the methodology employed in this study.

The overlapping time spans of the NDVI, LST, GRACE-S, and GPS-VD during 2012–2014 are employed to correlate the observed *R*, because the time series of all the remotely sensed variables share similar seasonal patterns to the observed *R*. However, notable differences are observed. For instance, both the GRACE-S and GPS-VD averaged at the upstream of the MRB against the *R* time series present a variable time lag over the entire time span (Figure 4). This can be attributable to a slightly different hydrological condition between the upstream and the downstream of the MRB each year under climatic variability (e.g., [98]). Regardless of averaging GPS-VD over the bounded square of the entire Yunnan Province or LRWY, GPS-VD yields an abnormal pattern between January and April of 2014. This can be due to GPS-VD being more sensitive to local hydrological variations or events when compared to GRACE-S [51].

To investigate the improvement resulting from the standardization, the observed *R* is standardized and compared to the standardized GRACE-S (hereinafter called GRACE-SI) and GPS-VD (hereinafter called GPS-DSI) using Equation (1). The above model fitting procedure is applied to correlate the standardized *R* with other standardized variables (i.e., PDSI, GRACE-SI, and GPS-DSI).

The estimated parameters and their corresponding standard deviation of Equation (2) is displayed in Table 1. Large offsets (i.e., *b*) and standard deviation are shown for the in situ *R* fitted with NDVI and LST. Hence, it is expected that the reconstructed *R* from NDVI and LST are not fitted well, as shown in Figure 5. The in situ *R* fitted with GRACE-S and GPS-VD appears to be better than that of NDVI and LST. This linear fitting results preliminarily indicate that our proposed space geodetic-observed variables (i.e., GRACE-S and GPS-VD) are better in *R* reconstruction.

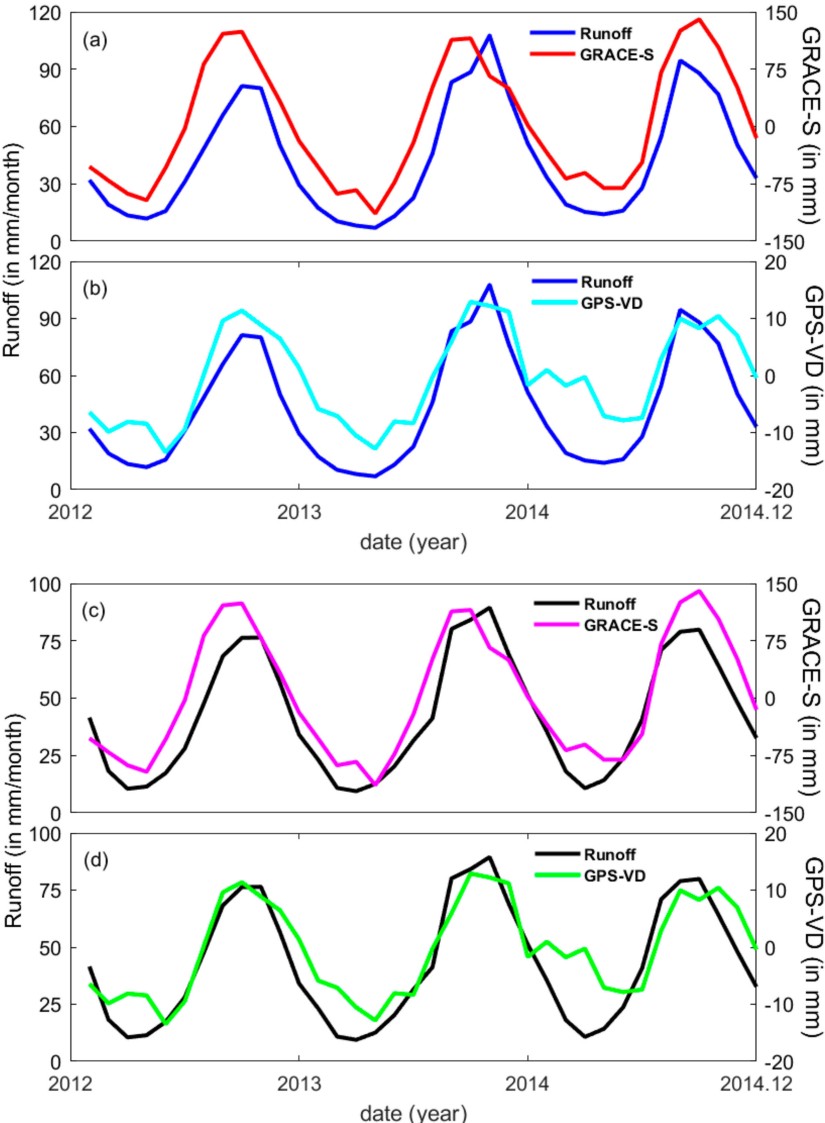

**Figure 4.** (**a**,**c**) Time series of runoff against averaged GRACE terrestrial water storage (GRACE-S) from CSR RL05 data and (**b**,**d**) GPS vertical displacement (GPS-VD) of the entire Yunnan Province at (top) MC-pair and (bottom) TC-pair stations.

### 4.2. Assessment Metrics

Three performance evaluation metrics, including the Pearson correlation coefficient (PCC), normalized root mean square error (NRMSE), and Nash–Sutcliffe efficiency (NSE) model coefficient, are employed to evaluate the reconstructed *R* against the observed *R* at in situ stations.

The PCC, which describes the degree of collinearity between two variables, is defined as follows:

$$PCC = \frac{\frac{1}{N}\sum_{i=1}^{N}\left(R_0(i) - \overline{R}_0\right)\left(R_e(i) - \overline{R}_e\right)}{\sqrt{\frac{1}{N}\sum_{i=1}^{N}\left(R_0(i) - \overline{R}_0\right)^2}\sqrt{\frac{1}{N}\sum_{i=1}^{N}\left(R_e(i) - \overline{R}_e\right)^2}} \qquad (3)$$

The NRMSE, being the RMSE divided by the maximum fluctuating range of observations, is defined as follows:

$$NRMSE = \frac{\sqrt{\frac{1}{N}\sum_{i=1}^{N}(R_e(i) - R_0(i))^2}}{\max(R_0) - \min(R_0)} \tag{4}$$

The NSE model coefficient, ranging from $-\infty$ to 1, is a performance indicator for evaluating the predictive power of the estimated $R$ compared to the in situ $R$ [99]. When the NSE model coefficient is closer to 1, the performance of the estimated $R$ is better. It is defined as follows:

$$NSE = 1 - \frac{\sum_{i=1}^{N}(R_e(i) - R_0(i))^2}{\sum_{i=1}^{N}(R_e(i) - \overline{R}_0)^2} \tag{5}$$

where $R_0(i)$ and $R_e(i)$ represent the observed and estimated $R$s of month $i$, respectively; $\overline{R}_0$ and $\overline{R}_e$ are the means of $R_0$ and $R_e$, respectively; $\max(R_0)$ and $\min(R_0)$ are the maximum and minimum $R$ of the in situ time series, respectively.

**Table 1.** Internal evaluation of runoff reconstructed at MT-pair station, and external evaluation of runoff estimated at TC-pair station based on reconstructed $R$ from relationships between MC-pair station and abovementioned variables for the entire Yunnan Province.

| Station | Data | Version | a (/Month) | b (mm/m) | Standard Deviation (a &b) | |
|---|---|---|---|---|---|---|
| MC pair reconstruction | Traditional RS data | NDVI | 0.0317 | −165.7979 | 0.0024 | 16.1466 |
| | | LST | −0.1313 | 1976.1705 | 0.0125 | 183.5319 |
| | Space geodetic-observed Variables | CSR RL05 | 0.3467 | 41.3759 | 0.0238 | 1.8711 |
| | | GFZ RL05 | 0.3419 | 41.2598 | 0.0233 | 1.8605 |
| | | JPL RL05 | 0.3515 | 43.1586 | 0.0221 | 1.7327 |
| | | CSR-mascon | 0.2797 | 41.8492 | 0.0177 | 1.7470 |
| | | CSR RL06 | 0.3419 | 41.3328 | 0.0214 | 1.7252 |
| | | GPS-VD | 4.3926 | 45.4435 | 0.2740 | 1.7290 |
| TC pair reconstruction | Traditional RS data | NDVI | 0.0281 | −142.1093 | 0.0020 | 13.3132 |
| | | LST | −0.1143 | 1725.1072 | 0.0111 | 163.8818 |
| | Space geodetic-observed Variables | CSR RL05 | 0.3039 | 41.3581 | 0.0207 | 1.6265 |
| | | GFZ RL05 | 0.2994 | 41.2575 | 0.0204 | 1.6280 |
| | | JPL RL05 | 0.3087 | 42.9218 | 0.0189 | 1.4840 |
| | | CSR-mascon | 0.2419 | 41.7816 | 0.0169 | 1.6591 |
| | | CSR RL06 | 0.2984 | 41.3252 | 0.0192 | 1.5479 |
| | | GPS-VD | 3.7715 | 44.8722 | 0.2726 | 1.7202 |

## 5. Evaluation and Discussion

This section illustrates the accuracy performance of the $R$ time series' reconstruction and estimation for internal and external evaluations, respectively. These time series are from the traditional RSD (NDVI and LST), space geodetic-observed variables (GRACE-S and GPS-VD), and their corresponding standardizations (GRACE-SI and GPS-DSI), including the drought index (PDSI). The internal and external evaluations of results are both applied to the MC-pair and TC-pair stations, so that the estimated $R$ from these two pairs can be compared against each other to assess the residual ocean tidal effect at the estuary. Note that the MC-pair station is closer to the estuary mouth than the TC-pair station with the former being located ~100 km and the latter ~220 km from the estuary mouth. The combined internal and external evaluations could quantify a portion of the systematic ocean tidal backwater effect on both reconstructed and estimated $R$s.

Serving as baseline results, both NDVI- and LST-reconstructed $R$s from the LRWY exhibit temporal patterns similar to the observed $R$s from both the MC-pair (Figure 5a,b) and TC-pair (Figure 5c,d) stations. However, relatively large differences are presented in peaks and troughs for the NDVI- and LST-reconstructed $R$s when compared to against the in situ $R$ (Figure 5c,d). While no apparent time lag for the NDVI-reconstructed $R$ against the in situ $R$, the in situ $R$ is lagged behind the LST-reconstructed $R$ from March 2013 to September 2014. We speculate that substantial differences between meteorological conditions (e.g., LST) at the upstream and hydrological conditions (e.g., $R$) at the downstream of the MRB (i.e., MRD) might exist during the above period. Note also that LST is the localized RS quantity that has no direct relationship with the $R$, as mentioned in the introduction of this study. This is the reason that it is served as a baseline result.

The $R$ reconstructions from GRACE-S and GPS-VD exhibit better results than those of the NDVI and LST-reconstructed $R$s (Figure 6) (Table 2), whereas their respective standardizations demonstrate even better performances (Figure 7) in capturing the peaks and troughs because a portion of the systematic effects is reduced by the standardization process. A similar situation is also observed in the TC-pair station time series with further reductions in the differences in peaks and troughs (Figure 7d–f), as consistently shown by the increase in PCC and NSE values of the reconstructed $R$ in the downstream MRB (Table 3). The PDSI-reconstructed $R$ exhibits results that are comparable to those of GRACE-SI and GPS-DSI because the PDSI is a hydrometeorological index generated from temperature and precipitation that captures the relative dryness of river basins in relation to river discharge variations [100]. Among all GRACE-reconstructed $R$s, the one reconstructed from JPL RL05 yields the best result. The one reconstructed from RL05 and RL06 of CSR show similar performances, whereas that of CSR-mascon solution yields a comparable performance.

By evaluating the differences of the MC-pair and TC-pair estimations in the assessment metrics between Tables 2 and 3 or between Tables 4 and 5, it is found that the usage of TC-pair account for a 1–3% decrease in the relative error, when compared to that using MC-pair close to the estuary mouth. This is the remaining backwater effect due to ocean tides of the MC-pair in the estuary. The standardization process yields a 2–7% increase in accuracy, no matter which pair station is used. The $R$ reconstructed from the GPS-DSI yields the lowest NRMSE value when accounting for the LRWY of the upstream MRB only, indicating that it remains subject to the total relative error of less than 9%. It is speculated that the remaining errors may be caused by our methodology, remaining environmental signals in the data time series, and potential time lag of less than a month between the upstream MRB and the MRD.

Overall, the estimated $R$s are slightly less accurate than the reconstructed $R$s, whereas their relative rankings remain practically the same. This could be partially caused by internal errors that are introduced by the reverse process. The proposed methodology that employs the upstream standardized GPS-VD (i.e., GPS-DSI) is proven to be a viable alternative to the estimation of $R$ in the MRD located at the estuary mouth of the MRB. However, the limitations of this study are that one in situ discharge time series in the river delta or estuary is required, and the GPS stations should be situated on the bedrock surface for observing the elastic deformation due to seasonal water storage changes.

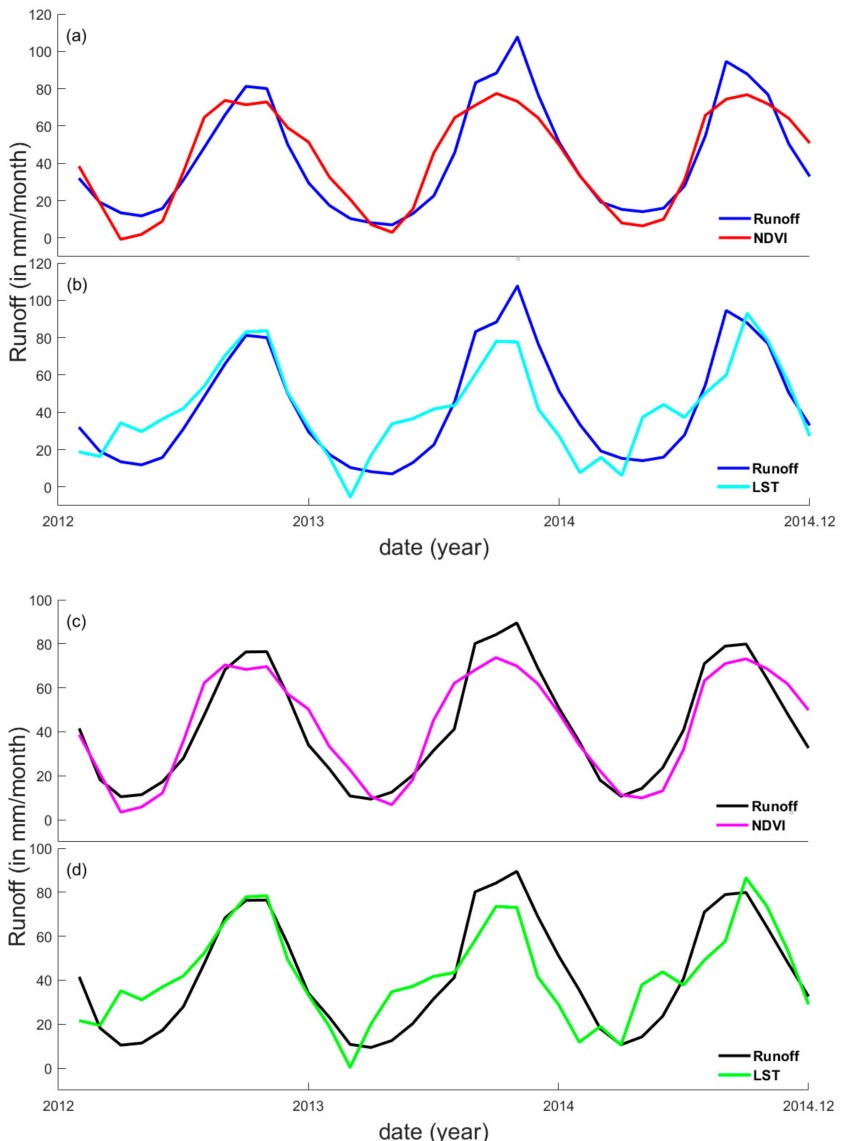

**Figure 5.** Runoff reconstructed from (**a**,**c**) NDVI and (**b**,**d**) LST at (**a**,**b**) MC-pair station, and (**c**,**d**) TC-pair station.

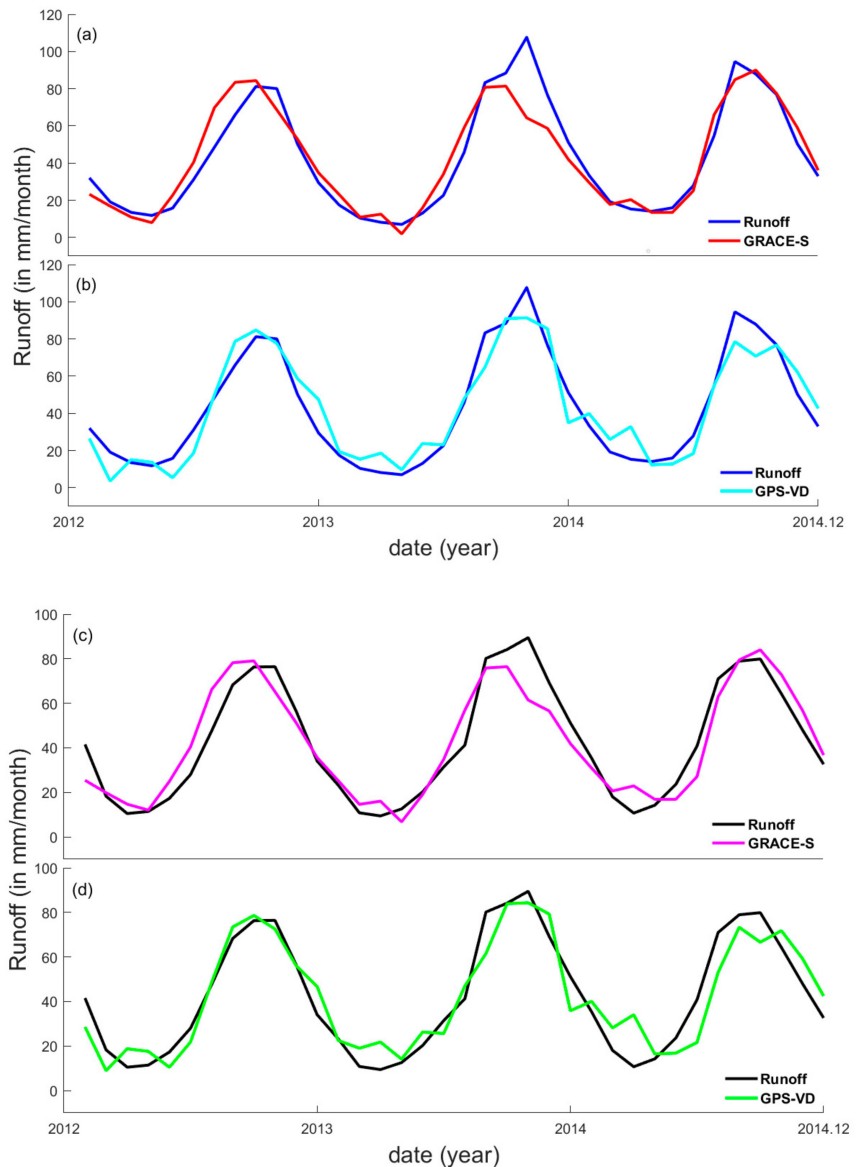

**Figure 6.** Runoff reconstructed from (**a**,**c**) GRACE and (**b**,**d**) GPS at (**a**,**b**) MC-pair station, and (**c**,**d**) TC-pair station.

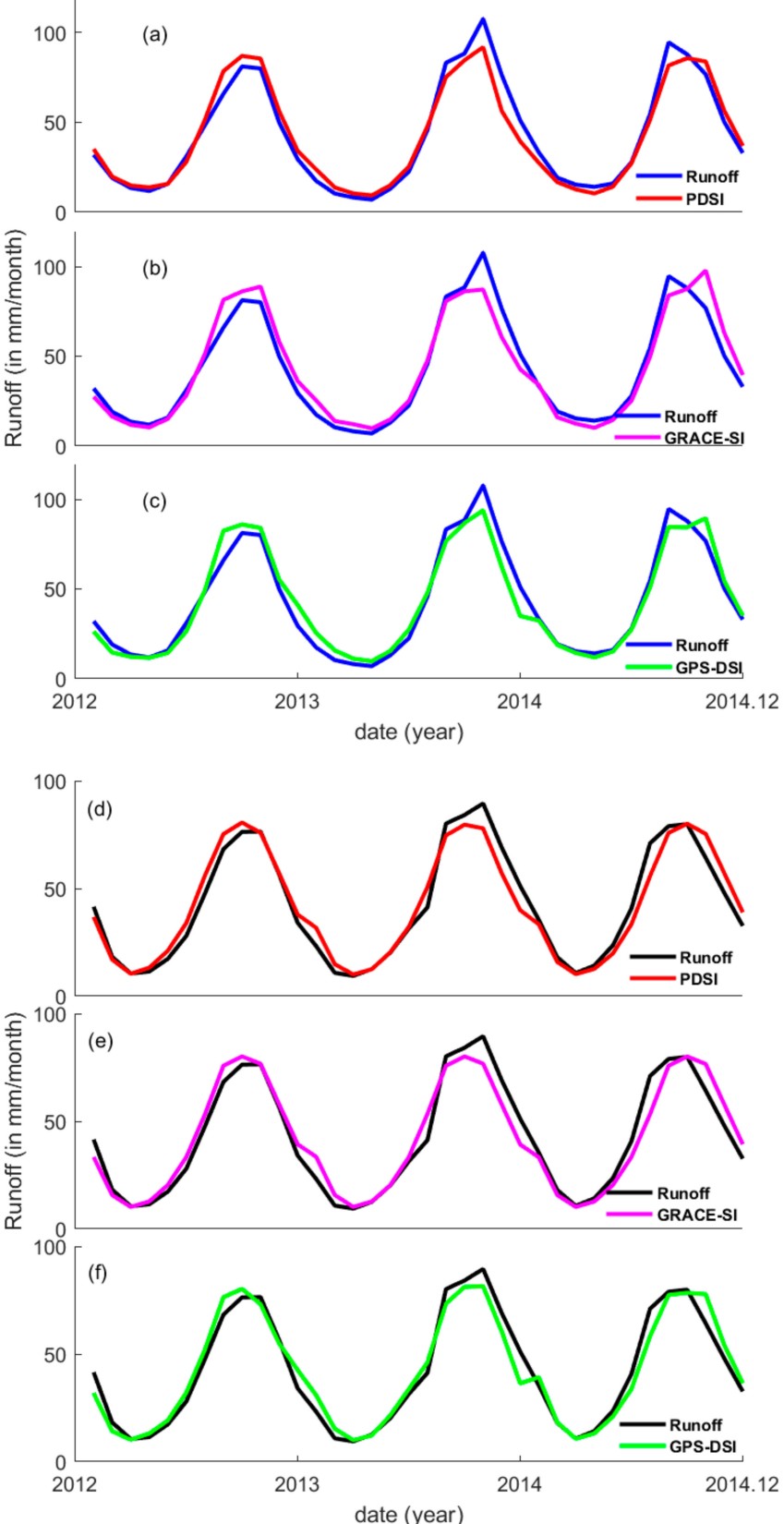

**Figure 7.** Runoff reconstructed from (**a**,**d**), PDSI, (**b**,**e**) GRACE-SI, and (**c**,**f**) GPS-DSI, at (**a**–**c**) MC-pair station and (**d**–**f**) TC-pair station.

**Table 2.** Internal evaluation of runoff reconstructed at MT-pair station, and external evaluation of runoff estimated at TC-pair station based on reconstructed *R* from relationships between MC-pair station and abovementioned variables for the entire Yunnan Province.

| Station | Variables/Indices | | PCC | NRMSE | NSE |
|---|---|---|---|---|---|
| MC-pair reconstruction | Traditional RS data | NDVI | 0.913 | 0.119 | 0.833 |
| | | LST | 0.875 | 0.141 | 0.766 |
| | Space geodetic-observed Variables | CSR RL05 | 0.928 | 0.108 | 0.862 |
| | | GFZ RL05 | 0.929 | 0.108 | 0.864 |
| | | JPL RL05 | 0.939 | 0.100 | 0.882 |
| | | CSR-mascon | 0.938 | 0.101 | 0.880 |
| | | CSR RL06 | 0.940 | 0.100 | 0.883 |
| | | GPS-VD | 0.940 | 0.100 | 0.883 |
| | Drought Indices | CSR RL05 | 0.963 | 0.079 | 0.927 |
| | | GFZ RL05 | 0.963 | 0.079 | 0.927 |
| | | JPL RL05 | 0.971 | 0.070 | 0.942 |
| | | CSR-mascon | 0.962 | 0.080 | 0.925 |
| | | CSR RL06 | 0.966 | 0.076 | 0.933 |
| | | PDSI | 0.984 | 0.053 | 0.967 |
| | | GPS-DSI | 0.970 | 0.070 | 0.942 |
| TC-pair estimated from MC-pair reconstruction | Traditional RS data | NDVI | 0.923 | 0.129 | 0.837 |
| | | LST | 0.870 | 0.164 | 0.739 |
| | Space geodetic-observed Variables | CSR RL05 | 0.930 | 0.126 | 0.847 |
| | | GFZ RL05 | 0.929 | 0.126 | 0.846 |
| | | JPL RL05 | 0.942 | 0.116 | 0.870 |
| | | CSR-mascon | 0.926 | 0.129 | 0.837 |
| | | CSR RL06 | 0.936 | 0.121 | 0.858 |
| | | GPS-VD | 0.922 | 0.134 | 0.826 |
| | Drought Indices | CSR RL05 | 0.948 | 0.118 | 0.865 |
| | | GFZ RL05 | 0.947 | 0.120 | 0.861 |
| | | JPL RL05 | 0.955 | 0.111 | 0.880 |
| | | CSR-mascon | 0.946 | 0.118 | 0.864 |
| | | CSR RL06 | 0.952 | 0.113 | 0.877 |
| | | PDSI | 0.967 | 0.091 | 0.920 |
| | | GPS-DSI | 0.957 | 0.103 | 0.896 |

**Table 3.** Internal evaluation of runoff reconstructed at TC-pair station, and external evaluation of runoff estimated at MC-pair station based on reconstructed *R* from relationships between TC-pair station and abovementioned variables for the entire Yunnan Province.

| Station | Variables/Indices | | PCC | NRMSE | NSE |
|---|---|---|---|---|---|
| TC-pair reconstruction | Traditional RS data | NDVI | 0.923 | 0.124 | 0.852 |
| | | LST | 0.870 | 0.159 | 0.756 |
| | Space geodetic-observed Variables | CSR RL05 | 0.930 | 0.118 | 0.864 |
| | | GFZ RL05 | 0.929 | 0.118 | 0.864 |
| | | JPL RL05 | 0.942 | 0.108 | 0.887 |
| | | CSR-mascon | 0.926 | 0.121 | 0.858 |
| | | CSR RL06 | 0.936 | 0.113 | 0.877 |
| | | GPS-VD | 0.922 | 0.125 | 0.849 |
| | Drought Indices | CSR RL05 | 0.964 | 0.086 | 0.929 |
| | | GFZ RL05 | 0.964 | 0.086 | 0.929 |
| | | JPL RL05 | 0.964 | 0.085 | 0.929 |
| | | CSR-mascon | 0.964 | 0.086 | 0.929 |
| | | CSR RL06 | 0.962 | 0.087 | 0.926 |
| | | PDSI | 0.975 | 0.071 | 0.951 |
| | | GPS-DSI | 0.969 | 0.080 | 0.938 |
| MC-pair estimated from TC-pair reconstruction | Traditional RS data | NDVI | 0.913 | 0.123 | 0.822 |
| | | LST | 0.875 | 0.145 | 0.753 |
| | Space geodetic-observed Variables | CSR RL05 | 0.928 | 0.113 | 0.849 |
| | | GFZ RL05 | 0.929 | 0.113 | 0.850 |
| | | JPL RL05 | 0.939 | 0.106 | 0.868 |
| | | CSR-mascon | 0.938 | 0.108 | 0.864 |
| | | CSR RL06 | 0.940 | 0.106 | 0.868 |
| | | GPS-VD | 0.940 | 0.107 | 0.866 |
| | Drought Indices | CSR RL05 | 0.955 | 0.093 | 0.898 |
| | | GFZ RL05 | 0.956 | 0.092 | 0.901 |
| | | JPL RL05 | 0.956 | 0.092 | 0.901 |
| | | CSR-mascon | 0.953 | 0.095 | 0.894 |
| | | CSR RL06 | 0.954 | 0.094 | 0.896 |
| | | PDSI | 0.970 | 0.080 | 0.924 |
| | | GPS-DSI | 0.960 | 0.089 | 0.907 |

**Table 4.** Internal evaluation of runoff reconstructed at MT-pair station, and external evaluation of runoff estimated at TC-pair station based on reconstructed *R* from relationships between MC-pair station and abovementioned variables for Lancang River within Yunnan Province.

| Station | Variables/Indices | | PCC | NRMSE | NSE |
|---|---|---|---|---|---|
| MC-pair reconstruction | Traditional RS data | NDVI | 0.905 | 0.124 | 0.818 |
| | | LST | 0.817 | 0.168 | 0.667 |
| | Space geodetic-observed Variables | CSR RL05 | 0.954 | 0.088 | 0.910 |
| | | GFZ RL05 | 0.956 | 0.085 | 0.915 |
| | | JPL RL05 | 0.958 | 0.084 | 0.918 |
| | | CSR-mascon | 0.918 | 0.116 | 0.843 |
| | | CSR RL06 | 0.946 | 0.094 | 0.895 |
| | | GPS-VD | 0.911 | 0.120 | 0.829 |
| | Drought Indices | CSR RL05 | 0.972 | 0.069 | 0.944 |
| | | GFZ RL05 | 0.970 | 0.072 | 0.940 |
| | | JPL RL05 | 0.984 | 0.052 | 0.969 |
| | | CSR-mascon | 0.967 | 0.075 | 0.935 |
| | | CSR RL06 | 0.971 | 0.070 | 0.942 |
| | | PDSI | 0.974 | 0.067 | 0.948 |
| | | GPS-DSI | 0.972 | 0.069 | 0.945 |
| TC-pair estimated from MC-pair reconstruction | Traditional RS data | NDVI | 0.929 | 0.124 | 0.851 |
| | | LST | 0.816 | 0.189 | 0.652 |
| | Space geodetic-observed Variables | CSR RL05 | 0.945 | 0.114 | 0.873 |
| | | GFZ RL05 | 0.947 | 0.113 | 0.876 |
| | | JPL RL05 | 0.952 | 0.108 | 0.887 |
| | | CSR-mascon | 0.897 | 0.150 | 0.782 |
| | | CSR RL06 | 0.937 | 0.121 | 0.858 |
| | | GPS-VD | 0.888 | 0.156 | 0.764 |
| | Drought Indices | CSR RL05 | 0.955 | 0.111 | 0.881 |
| | | GFZ RL05 | 0.950 | 0.118 | 0.866 |
| | | JPL RL05 | 0.966 | 0.099 | 0.905 |
| | | CSR-mascon | 0.949 | 0.113 | 0.875 |
| | | CSR RL06 | 0.957 | 0.106 | 0.891 |
| | | PDSI | 0.962 | 0.096 | 0.911 |
| | | GPS-DSI | 0.956 | 0.105 | 0.893 |

**Table 5.** Internal evaluation of runoff reconstructed at TC-pair station, and external evaluation of runoff estimated at MC-pair station based on reconstructed *R* from the relationships between TC-pair station and abovementioned variables for Lancang River within Yunnan Province.

| Station | Variables/Indices | | PCC | NRMSE | NSE |
|---|---|---|---|---|---|
| TC-pair reconstruction | Traditional RS data | NDVI | 0.929 | 0.119 | 0.862 |
| | | LST | 0.816 | 0.186 | 0.666 |
| | Space geodetic-observed Variables | CSR RL05 | 0.945 | 0.105 | 0.894 |
| | | GFZ RL05 | 0.947 | 0.103 | 0.897 |
| | | JPL RL05 | 0.952 | 0.098 | 0.907 |
| | | CSR-mascon | 0.897 | 0.142 | 0.805 |
| | | CSR RL06 | 0.937 | 0.112 | 0.879 |
| | | GPS-VD | 0.888 | 0.148 | 0.788 |
| | Drought Indices | CSR RL05 | 0.965 | 0.085 | 0.930 |
| | | GFZ RL05 | 0.964 | 0.085 | 0.930 |
| | | JPL RL05 | 0.970 | 0.078 | 0.941 |
| | | CSR-mascon | 0.963 | 0.087 | 0.927 |
| | | CSR RL06 | 0.963 | 0.087 | 0.927 |
| | | PDSI | 0.969 | 0.079 | 0.940 |
| | | GPS-DSI | 0.972 | 0.076 | 0.945 |
| MC-pair estimated from TC-pair reconstruction | Traditional RS data | NDVI | 0.905 | 0.127 | 0.810 |
| | | LST | 0.817 | 0.171 | 0.656 |
| | Space geodetic-observed Variables | CSR RL05 | 0.954 | 0.095 | 0.894 |
| | | GFZ RL05 | 0.956 | 0.093 | 0.899 |
| | | JPL RL05 | 0.958 | 0.091 | 0.902 |
| | | CSR-mascon | 0.918 | 0.122 | 0.825 |
| | | CSR RL06 | 0.946 | 0.101 | 0.880 |
| | | GPS-VD | 0.911 | 0.127 | 0.812 |
| | Drought Indices | CSR RL05 | 0.957 | 0.091 | 0.904 |
| | | GFZ RL05 | 0.957 | 0.091 | 0.903 |
| | | JPL RL05 | 0.965 | 0.083 | 0.918 |
| | | CSR-mascon | 0.954 | 0.094 | 0.897 |
| | | CSR RL06 | 0.955 | 0.093 | 0.899 |
| | | PDSI | 0.959 | 0.089 | 0.906 |
| | | GPS-DSI | 0.963 | 0.086 | 0.913 |

## 6. Conclusions

In lieu of employing traditional remote sensing (RS) data for surface runoff (*R*) reconstruction, the potential use of upstream GPS vertical displacement (GPS-VD) and its standardization (GPS-DSI) for *R* reconstruction at estuaries on a monthly temporal scale is explored. It is found that the reconstructed *R* time series at the Mekong River Delta (MRD) from the Mekong River Basin (MRB) upstream GPS-VD are comparable to those from the GRACE terrestrial water storage (GRACE-S) and traditional RS data.

All reconstructed *R* time series from the standardized variables, including GRACE-SI, GPS-DSI, and PDSI, are found to have a 2–7% increase in accuracy in terms of the NRMSE when compared to those without standardization. The reconstructed *R* time series from the spatially averaged of the upstream GPS-DSI is shown to be comparable to that reconstructed by the PDSI, but better than those obtained by traditional RS data and GRACE-S. The internal evaluation also demonstrates that the reconstructed *R* based on GPS-DSI attains a PCC of 0.97 and NSE of 0.94 for both MC-pair and TC-pair stations. Despite being slightly less accurate than the reconstructed *R*, the estimated *R* exhibits an accuracy that is similar to the above as externally validated via another station location.

The comparison of *R* reconstruction and estimation from the MC-pair and TC-pair stations indicates that the remaining backwater effect induced by ocean tides yields a1%–3% in the relative

error on the estimated *R*s in this study. The *R* reconstructed from the GPS-DSI yields the lowest NRMSE value of less than 9% when accounting for the main upstream area of the MRB (i.e., Lancang River within Yunnan Province). This reveals that the best reconstructed *R* from the GPS-DSI remains subject to the total relative error of ~9%. This may be caused by our methodology, the remaining environmental signals in the data time series, and the potential time lag (less than a month) between the upstream MRB and the MRD.

Overall, the proposed methodology, which employs the upstream GPS-VD and its standardization, is proven to be a potential alternative for reconstructing and estimating *R* in the MRB. It is anticipated that the proposed GPS-VD and its standardization can also be applied to basin-wide discharge estimations and potentially replace the function of GRACE-S in terms of water balance. Higher temporal resolutions of the reconstructed and estimated *R*s can also be achieved via the GPS because the use of daily GPS-VD solutions have become a standard practice, and also offers acceptable accuracy.

**Author Contributions:** H.S.F. initiated the experimental design, performed data collection, conducted data pre-processing and result interpretation, and wrote the manuscript. L.Z. conducted an entire experiment, including post-processing of remotely sensed data. Y.L. performed GPS data pre-processing. Z.M. visualized data. Y.C. proofread the revised manuscript and provided critical comments. All authors have read and agreed to the published version of the manuscript.

**Funding:** This study was financially supported by the National Natural Science Foundation of China (NSFC) Grants No.: 41674007, 41974003, 41374010, and 41429401.

**Acknowledgments:** We purchased the discharge data from the Mekong River Commission (MRC), financially supported by NSFC Grant No.: 41374010.

**Conflicts of Interest:** The authors declare no conflicts of interest.

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
