# Peer review of "Upstream GPS Vertical Displacement and its Standardization for Mekong River Basin Surface Runoff Reconstruction and Estimation"

_remotesensing, doi:10.3390/rs12010018_

Round 1

Reviewer 1 Report

Title: Upstream GPS vertical Displacement and its Standardization for River Basin Surface Runoff Reconstruction and Estimation.

In this paper, the authors propose a methodology that uses upstream GPS Vertical Displacement and its standardization to reconstructing and estimating surface runoff.

This is an interesting study. However, concerning the methodology and the discussion of results, I think the paper is incomplete. Then I cannot recommend this paper for publication in remote sensing. But I will consider my decision if the authors improve the discussion.

Main Suggestions:

Section 3.3: Why do the authors use the CSR RL05 solution? Why don’t they use the GFZ, JPL or mascons solutions?

Section 4:

Please, could the authors comment the figures 4, 5 and 6? These figures are not enough commented.  It seems to be lags and larger discrepancies between the series. Could they explain the origin?

In figures 5 and 6, PSDI, GRACE-SI, and GPS DSI are equal for the MC- pair, and TC-pair. Then, why is it interesting to analyze them separately? Then, can the MC- pair, and TC-pair be used for the external and the internal evaluation?

Line 254-256: Please, could the authors explain the reconstruction and estimation procedures?

Please, could the authors give the accuracy of the parameters determined from the reconstructed R?

Section5:

Figure 7: It seems to exist a time lag between the series. Also, we can find a nearly semi-annual signal in the reconstructed runoff based on LST. Could the authors explain these facts?  

Also in figure 7, the reconstructed runoff based on NDVI and the reconstructed runoff based on LST for MC-pair and TC-pair stations, respectively, are the same, why?

Line 307-309: The authors wrote: “ Notice that the process of summing up the My Thuan and Cantho station data time series should reduce the seasonal tidal backwater effect during the data processing steps.” Could they clarify this point?

Line 316-320: The authors wrote “By evaluating the differences in the assessment metrics between Tables 1 and 2 or between Tables 3 and 4, it is found that the backwater effect and standardization process account for a 1–3% decrease in the relative error and a 2–7% increase in accuracy, respectively.” Is this improvement really significant?

Other suggestions:

Line 62-63: Please, could the authors add a reference concerning the correlation between river water discharge, altimetry and GRACE observations.

Figure 4: Please, the authors should add the units of GRACE-S and GPS-VD.

General suggestion: I think there are too many abbreviations.

Author Response

Please read the response in the attached PDF.

Reviewer 2 Report

General comments

The article gives a comprehensive and clear description of the upstream GPS vertical displacement and its standardization for river basin surface runoff reconstruction and estimation. The analysis performed for the Mekong River Basin and Mekong River Delta is based on 1) Hydrological Stations and Passive Remote Sensing Data, 2) Palmer Drought Severity Index, 3) GRACE Release-05 monthly solutions provided by the University of Texas, Centre for Space Research (CSR), 4) Vertical Displacements determined form GPS observations from 33 GPS stations in Yunnan Province using GAMIT FORTRAN software. The concern of the reviewer is why the authors did not use GRACE Release-06 monthly solutions (see line 202, page 5), available for long time already, which provide more reliable interpretation of the observed signal. Also it is not clear whether the authors really mean GPS observations in the manuscript since the majority of stations acquiring data from satellite navigation systems are nowadays equipped with the receiver/antenna systems tracking besides GPS other GNSS signals like e.g. GLONASS, BeiDou etc.

Detail comment

page 5, line 202 – The use of Gaussian filter with a 350 km radius in a de-stripping procedure should be explained. Why Gaussian filter and why 350 radius?

The manuscript needs minor improvement. The improved version of the manuscript (considering reviewer's comments) can be recommended for publication.

Author Response

The article gives a comprehensive and clear description of the upstream GPS vertical displacement and its standardization for river basin surface runoff reconstruction and estimation. The analysis performed for the Mekong River Basin and Mekong River Delta is based on 1) Hydrological Stations and Passive Remote Sensing Data, 2) Palmer Drought Severity Index, 3) GRACE Release-05 monthly solutions provided by the University of Texas, Centre for Space Research (CSR), 4) Vertical Displacements determined form GPS observations from 33 GPS stations in Yunnan Province using GAMIT FORTRAN software.

Response: Thank you.

The concern of the reviewer is why the authors did not use GRACE Release-06 monthly solutions (see line 202, page 5), available for long time already, which provide more reliable interpretation of the observed signal.

Response: Most previous research studies employed GRACE Release-05 monthly solutions. To be conservative, we did not use Release-06 monthly solutions. But given your comment and other reviewers' suggestions, we further used CSR GRACE Release-06 and mascon monthly solutions to derive the runoff. All texts are updated to include the results based on popular GRACE solutions in the revised manuscript accordingly. Please read the evaluation statistics in the tables in the revised manuscript.

Also it is not clear whether the authors really mean GPS observations in the manuscript since the majority of stations acquiring data from satellite navigation systems are nowadays equipped with the receiver/antenna systems tracking besides GPS other GNSS signals like e.g. GLONASS, BeiDou etc.

Response: We mean GPS observations only. Perhaps you are right that GLONASS option may be presented. We just processed GPS signal for the GPS vertical displacement only. The background is that most continuous GPS stations from the Crustal Movement Observation Network of China (CMONOC) in the study region have started continuous observations since 2010 (i.e., CMONOC phase 2 installed in year 2008), except KMIN and XIAG stations, which have been operated since 1999 (i.e., CMONOC phase 1). Note that only 14 Beidou satellites have been launched at the end of year 2012. Therefore, the antennas equipped with observation ability for other GNSS signals like BeiDou seem not possible at the time of installation, as checked.

Detail comment

page 5, line 202 – The use of Gaussian filter with a 350 km radius in a de-stripping procedure should be explained. Why Gaussian filter and why 350 radius?

Response: The de-stripping procedure and Gaussian filter were performed separately. It should be the English expression that misleads the reviewer. We will revise the text accordingly to convey our meaning accurately as below:

"In addition, a de-striping procedure is applied. As tested in this study, a Gaussian filtering with a 350-km radius is the optimal radius chosen to reduce the uncertainties arising from correlated errors of TWS data in space at finer resolutions [87,88]."

Gaussian filtering is a conventional method to reduce the spatially correlated errors. Because the aim of this manuscript is to demonstrate upstream GPS vertical displacements in deriving downstream runoff while GRACE is served as a baseline indicator (but NOT the main objective), other filters are not tested. Furthermore, as shown in our result from the tables, most results achieved a very good PCC, NSE, and NRMSE already. Using other filtering techniques may not substantially further improve the evaluation statistics.

In essence, we tested the radius of 250-km, 300-km, 350-km, and 400-km of the Gaussian filtering, we empirically found that 350-km radius is the optimal one in our study region. Therefore, we used it.

The manuscript needs minor improvement. The improved version of the manuscript (considering reviewer's comments) can be recommended for publication.

Response: We hope that the above responses and the revised manuscript can satisfy with your above requirements.

Reviewer 3 Report

Dear Authors,

Upstream GPS Vertical Displacement and its Standardization for River Basin Surface Runoff Reconstruction and Estimation

This paper is well written and very interesting. But like in any study in this area using GPS vertical Displacement, maybe the author could include the limitations of the study, especially for basins with few gauging information? Also, would similar conclusion be reached if this is done in another basin with different physiographic characteristics?

Specifically, I have the following comments:

In Figure 1. Where is Lancang River Basin?

Line 115-116: Section 6 summarizes the conclusions. There is no Section 6. The manuscript needs re-arrangement.

Line 34-35: I found this difficult to follow: "The reconstructed R from the standardized GPS-VD yields the lowest NRMSE (total relative error < 9%) when it is taken into account only for the Lancang River basin of the upstream MRB".

Author Response

This paper is well written and very interesting.

Response: Thank you.

But like in any study in this area using GPS vertical Displacement, maybe the author could include the limitations of the study, especially for basins with few gauging information?

Response: We wrote the sentences related to the limitations of this study in the last paragraph of "Section 5 Evaluation and Discussion" as below:

"However, the limitations of this study are that one in-situ discharge time series in the river delta or estuary is required, and the GPS stations should be situated on the bedrock surface for observing the elastic deformation due to seasonal water storage changes."

Also, would similar conclusion be reached if this is done in another basin with different physiographic characteristics?

Response: No one can guarantee the presented methodology should work for all basins with different physiographic characteristics. That is why we did not give this strong statement in the conclusion of the manuscript. But we anticipate similar conclusion would be reached for large-scale river basin, such as Yangtze River, Pearl River, etc.

Specifically, I have the following comments:
In Figure 1. Where is Lancang River Basin?

Response: In essence, Lancang River Basin is the upstream of Mekong River Basin from headwater source region to southern exit of Yunnan province. Based on your comment, we re-plot the geographic region so that the headwater source region can be seen in the revised manuscript.

Line 115-116: Section 6 summarizes the conclusions. There is no Section 6. The manuscript needs re-arrangement.

Response: Yes, we did the correction. "5. Conclusions" should be "6. Conclusions".

Line 34-35: I found this difficult to follow: "The reconstructed R from the standardized GPS-VD yields the lowest NRMSE (total relative error < 9%) when it is taken into account only for the Lancang River basin of the upstream MRB".

Response: Thank you for pointing it out. It is apparent that the above expression is not easy to understand. We re-write it below:

"The reconstructed R from the standardized GPS-VD yields the lowest total relative error of less than 9% when accounting for the main upstream area of the MRB."

Based on your comment, we re-checked every English sentence and expression and mistakes again in the revised manuscript. Thank you for your careful reading.

Round 2

Reviewer 1 Report

Title: Upstream GPS vertical Displacement and its Standardization for River Basin Surface Runoff Reconstruction and Estimation.

In this paper, the authors propose a methodology that uses upstream GPS Vertical Displacement and its standardization to reconstructing and estimating surface runoff.

This is an interesting study. The paper has been largely improved and completed. The authors have answered my questions and they have taken into account my suggestions.  I recommend now this paper for publication in Remote Sensing journal.

However, I have some minor suggestions.

Section 3.3 Description of GRACE solution is not clear. Please, could the authors revise this paragraph?

Figure 3,4: The graphics should be on the same page.

Author Response

Please find the attached PDF for the reply with equation symbols that cannot be shown in text.

Title: Upstream GPS vertical Displacement and its Standardization for River Basin Surface Runoff Reconstruction and Estimation.

In this paper, the authors propose a methodology that uses upstream GPS Vertical Displacement and its standardization to reconstructing and estimating surface runoff.

This is an interesting study. The paper has been largely improved and completed. The authors have answered my questions and they have taken into account my suggestions.  I recommend now this paper for publication in Remote Sensing journal.

Response: Thank you.

However, I have some minor suggestions.

Section 3.3 Description of GRACE solution is not clear. Please, could the authors revise this paragraph?

Response: Yes, you are right. We messed up the difference between CSR-mascon and other GRACE level-2 data products. We revised it as below:

"Five monthly GRACE solution data products are employed, including the Center for Space Research (CSR) Release (RL) 05 (hereinafter abbreviated as CSR RL05), RL 06 (hereinafter abbreviated as CSR RL06), and its RL06 mascon solution (hereinafter abbreviated as CSR-mascon), Jet Propulsion Laboratory (JPL) RL 05 (hereinafter abbreviated as JPL RL05), and GeoforschungsZentrum (GFZ) RL 05 (hereinafter abbreviated as GFZ RL05). Except for the monthly CSR-mascon GRACE-S readily available at a 0.25°×0.25° grid that can be downloaded at http://www2.csr.utexas.edu/grace/RL06_mascons.html, all other monthly GRACE Level-2 data represent mass changes in terms of Stokes coefficients (SC) which can be downloaded at http://icgem.gfz-potsdam.de/series.

While the SC of the JPL RL 05 are expanded up to degree 90, the SC of the CSR RL05, CSR RL06, and GFZ RL05 are expanded up to degree 60. Using equations in [84], the SC of the CSR RL05, CSR RL06, GFZ RL05 and JPL RL05 can be converted into GRACE-S that is interpolated into a 1°×1° grid.

         Except for the CSR-mascon, before converting SC into GRACE-S time series, the degree-one and  terms in SC are added and replaced, respectively, to correct the geocenter motion and the geoid. [85,86]. In addition, a de-striping procedure is applied. As tested in this study, a Gaussian filtering with a 350-km radius is the optimal radius chosen to reduce the uncertainties arising from correlated errors of TWS data in space at finer resolutions [87,88]."

Figure 3,4: The graphics should be on the same page.

Response: Your concern should be the broken Figure 3 and Figure 4 (i.e. upper figure in the first page and lower figure in the second page). Yes, you are right. We aligned them accordingly in the revised manuscript. Final editing works should be left for MDPI staff. Thank you for pointing it out.
